# Roles of Two-Component Signal Transduction Systems in Shigella Virulence

**DOI:** 10.3390/biom12091321

**Published:** 2022-09-18

**Authors:** Martina Pasqua, Marco Coluccia, Yoko Eguchi, Toshihide Okajima, Milena Grossi, Gianni Prosseda, Ryutaro Utsumi, Bianca Colonna

**Affiliations:** 1Istituto Pasteur Italy, Department of Biology and Biotechnologies, Sapienza University of Rome, 00185 Rome, Italy; 2Faculty of Biology-Oriented Science and Technology, Kindai University, Kinokawa 649-6493, Wakayama, Japan; 3Sanken, The Institute of Scientific and Industrial Research, Osaka University, Ibaraki 567-0047, Osaka, Japan

**Keywords:** two-component signal transduction systems, *Shigella* virulence, bacterial regulation, stress response

## Abstract

Two-component signal transduction systems (TCSs) are widespread types of protein machinery, typically consisting of a histidine kinase membrane sensor and a cytoplasmic transcriptional regulator that can sense and respond to environmental signals. TCSs are responsible for modulating genes involved in a multitude of bacterial functions, including cell division, motility, differentiation, biofilm formation, antibiotic resistance, and virulence. Pathogenic bacteria exploit the capabilities of TCSs to reprogram gene expression according to the different niches they encounter during host infection. This review focuses on the role of TCSs in regulating the virulence phenotype of *Shigella*, an intracellular pathogen responsible for severe human enteric syndrome. The pathogenicity of *Shigella* is the result of the complex action of a wide number of virulence determinants located on the chromosome and on a large virulence plasmid. In particular, we will discuss how five TCSs, EnvZ/OmpR, CpxA/CpxR, ArcB/ArcA, PhoQ/PhoP, and EvgS/EvgA, contribute to linking environmental stimuli to the expression of genes related to virulence and fitness within the host. Considering the relevance of TCSs in the expression of virulence in pathogenic bacteria, the identification of drugs that inhibit TCS function may represent a promising approach to combat bacterial infections.

## 1. Introduction

Most pathogenic bacteria are able to survive in diverse environments where they continually face a variety of physical and chemical signals encountered both within or outside the host. The transition from a free-living state to a host-associated one usually requires major changes in growth conditions, e.g., pH, osmolarity, temperature, availability of oxygen, and nutrients. The ability of pathogenic bacteria to rapidly adapt to intracellular life depends on being able to sense environmental changes and respond appropriately, often with drastic changes in the cell’s transcriptional program. The capability to swiftly modulate gene expression requires bacteria to invest not only in gene functions that allow adaptation to different milieus but also in gene functions that coordinate the cell’s response to changing conditions. 

In this context, a central role is played by the so-called two-component signal transduction systems (TCS), which are able to integrate environmental signals with other metabolic and stress pathways, promoting the expression of genes required in a specific host niche [1,2,3,4] (Figure 1). TCSs are almost ubiquitous in bacteria, and several TCS are present in a cell’s genome. The important role played by TCSs in the adaptation of bacteria to the environment is further demonstrated by the fact that there is a relationship between the number of TCSs and genome size. While environmental bacteria with large genomes tend to have numerous TCSs, bacteria that have adapted to a particular niche within the host, including some pathogenic bacteria, tend to have a limited number of TCSs [1]. 

In bacterial cells, TCSs control a multitude of functions, including motility, biofilm formation, response to stress, and virulence. In this review, we discuss the contribution of TCSs to the modulation of genes.

## 2. The Pathogenicity Process in *Shigella*

*Shigella* is a highly adapted intracellular human pathogen, mainly found in the developing world and causing severe enteric syndrome. Shigellosis is extremely contagious and, although self-limiting, it may be fatal in immunocompromised persons and children [5,6]. *Shigella* has been traditionally divided into four subspecies (*S. flexneri*, *S. dysenteriae, S. sonnei*, and *S. boydii*); however, genomic analyses clearly show that *Shigella* belongs to the *Escherichia coli* species and has originated several times from different branches of the *E. coli* tree by an evolutionary process involving the gain and loss of genes [6,7,8]. The gain of the virulence plasmid (pINV) carrying the genes encoding a Type III secretion system (T3SS), the invasion plasmid antigens (Ipa proteins), and other determinants involved in bacterial survival and intra-/inter-cellular spread has been a major event towards a pathogenic lifestyle [9]. A relevant complementary step has been the inactivation of a series of genes that negatively interfere with the full expression of the virulence phenotype [10,11,12]. 

Severe inflammatory colitis caused by *Shigella* relies on the ability of the pathogen to invade colonocytes and spread among adjacent cells, eventually leading to the inflammatory destruction of the intestinal barrier function [13]. To enter its major cell target, once ingested, the *Shigella* must first endure the physiochemical barriers met while transiting through the digestive system. The bacteria must then cross the colonic epithelium by entering M cells and be delivered through transcytosis to the basolateral surface of the epithelium. Here, the bacteria come in contact with and are phagocytosed by resident macrophages and dendritic cells. The *Shigella* rapidly lyse the phagolysosome in a T3SS-dependent manner and, after a limited multiplication in the cytosol, escapes the macrophages by inducing inflammatory cell death (pyroptosis). Macrophage pyroptosis is accompanied by the secretion of the proinflammatory cytokines interleukine 1β (IL-1β) and IL-18 [14]. *Shigella* released by dying macrophages interacts with proteins at the basolateral side of colonocytes and promotes its uptake by inducing epithelial cell macropinocytosis. The vacuole is rapidly disrupted, freeing bacterial cells into the cytosol. Subsequently, the *Shigella* multiplies and, using actin-based motility, spreads to adjacent cells [13,15]. Invasion of and spread to neighboring epithelial cells as well as cytosolic replication of *Shigella* heavily depend on T3SS effectors as they facilitate primary and secondary vacuole rupture, favor cell-to-cell diffusion, damp the host inflammatory response, and promote host cell survival [16,17]. Altogether, these activities preserve the integrity of the epithelial replicative niche and favor conditions for further dissemination. The inflammatory response elicited in the host, although attenuated through the strategies devised by *Shigella*, ultimately leads to the clearance of invading bacteria mainly by neutrophil-mediated killing [13,14] (Figure 2). 

## 3. Regulation of *Shigella* Plasmid Virulence Genes

The pathogenicity of *Shigella* is the result of the complex action of a wide number of virulence determinants mainly located on the pINV genome [9,16,18]. The expression of pINV genes is controlled by multiple environmental stimuli through a regulatory cascade involving transcriptional regulators and sRNA encoded by both pINV and the chromosome [19]. Temperature is a crucial factor since transcription of the pINV virulence genes is strongly repressed at 30 °C. The primary event following entry into the host environment and the consequent upshift at the host temperature is the activation of the *virF* gene, which encodes an AraC-like transcription factor [20]. This triggers a cascade of events where VirF promotes the transcription of *icsA*, coding for an outer membrane protein involved in bacterial intra- and inter-cellular motility [21], and of *virB*, which encodes another transcriptional regulator [22]. Indeed, VirB activates several virulence genes, including those encoding the *Shigella* T3SS, its early effectors, and MxiE, the last regulator in the cascade. Finally, MxiE, whose correct reading frame is generated by transcriptional slippage, activates the late effector genes, assisted by the IpgC protein [15,19]. 

At low temperatures, the nucleoid-associated protein H-NS represses transcription of the virulence genes. In particular, H-NS is able to recognize and repress the *virF* promoter only below 32 °C [20]. This temperature dependence is due to the interaction of H-NS with two binding sites within the *virF* promoter, spaced by a DNA region endowed with sequence-mediated curvature. Increasing the temperature, the DNA curvature progressively relaxes, undermining the ability of H-NS to maintain a bridged structure that prevents access to the RNA polymerase [23]. Once VirF has been synthesized, it acts as an antisilencer, relieving H-NS repression at the *virB* [22] and *icsA* [24] promoters. Two other nucleoid-associated proteins, FIS and IHF, contribute to the regulation of the pINV virulence genes. FIS exerts direct positive control on *virF* by competing with H-NS for the binding at the promoter. The effect of FIS is particularly evident in the early exponential phase, in line with the growth phase-dependent expression of this protein, and likely favors an increased expression of *virF* once the bacteria gain access to the host environment [25]. Similarly to FIS, IHF exerts positive control on the *virF* and *virB* genes and also, in this case, activation is mediated by the direct binding of the IHF protein to the promoters [26]. The complex regulatory circuit controlling the expression of the *vir* genes also involves the activity of RnaG and RyhB, two regulatory sRNAs. RnaG is a pINV-encoded sRNA transcribed on the complementary strand of *icsA* [27]. RnaG negatively controls the expression of *icsA*, the gene responsible for actin-mediated motility of *Shigella*, by means of two independent mechanisms. In the first one, RnaG acts as antisense RNA and induces premature termination of the *icsA* transcript. In the second one, the high level of RnaG transcription interferes with the activity of *icsA* promoter [24]. RyhB is a chromosomally encoded sRNA produced under conditions of iron limitation. The RyhB-dependent regulation of the *Shigella* virulence phenotype is due to the repression of the *virB*, thus affecting many virulence genes, including those coding for the T3SS and their effectors [28]. Repression occurs at the transcriptional level since, in the presence of RyhB, a reduction in the steady state level of the *virB* transcript occurs. 

The regulation of the pINV virulence cascade is further complicated by the post-transcriptional modification of the *virF* transcript. On the one hand, the efficient translation of the *virF* mRNA requires tRNA modifications induced by TGT and Mia enzymes [29,30]. On the other hand, the presence of a different translation site and a leaderless mRNA causes *virF* transcription to generate a shorter form (VirF_21_) that acts as a repressor of the VirF expression itself [31,32]. The complex control exerted by chromosomal genes on virulence expression is exemplified not only by the involvement of the nucleoid-associated proteins H-NS, FIS, and IHF but also by the involvement of diverse TCSs, which deeply link environmental stimuli to the expression of genes related to virulence and fitness within the host, as we will highlight in the following sections. 

## 4. Two-Component Signal Transduction Systems of *Shigella*

To adapt to various environmental changes and survive, bacteria have TCSs (Figure 1), which regulate the expression of genes involved in cell growth and division, virulence, drug resistance, quorum sensing, biofilm formation, acid tolerance, spore formation, and nitrogen fixation [1,33,34]. TCSs are composed of sensors (histidine kinases; HKs) and response regulators (RRs). Most HKs are membrane-bound homodimeric proteins and are classified into two types depending on the combination of multiple domains (Figure 1). A typical HK consists of a sensor domain, a HAMP (or PAS) domain, a dimeric domain (DHp) containing a highly conserved His residue that is phosphorylated, and a catalytic domain (CA) containing an ATP-binding site (Figure 1a). In the DHp and CA domains, there are regions with highly conserved amino acid sequences: H-box, which contains the histidine phosphorylation site, in the DHp domain, and N, G1, F, and G2 boxes in the CA domain. Of the 29 HKs in the *E. coli* genome (Table 1), 23 have the domain organization shown in Figure 1a. When an HK senses an environmental signal, the signal is transmitted to the intracellular region as a conformational change of a transmembrane region. Finally, the ATP-bound CA domain approaches DHp so that the His residues can be phosphorylated (His-P) [3]. The phosphate group is transferred from His-P to a specific Asp residue conserved on the receiver domain (REC) of RR to regulate the effector domain (ED) function. Most EDs have DNA-binding ability, and RR-P acts as a transcriptional regulator to control the expression of genes [3]. 

Four of the 29 *E. coli* HKs (Table 1), TorS, ArcB, BarA, and EvgS, are hybrid sensors (Figure 1b) and contain HAMP (or PAS), DHp, CA, REC, and HPt domains (phosphate group-accepting domains different from DHp) (Figure 1b). First, DHp-His is autophosphorylated, and the phosphate is transferred via three steps, HK (DHp-His) → HK (REC-Asp) → HK (HPt-His) → RR (REC-Asp), resulting in the phosphorylation of RR (Figure 1b) [47]. Some HKs have phosphorylated histidine in their HPt (Histidine Phosphotransfer) domain (Class II HKs), whereas most HKs have conserved phosphorylated histidine in their DHp domain (Class I HK) [46]. 

HKs and RRs predicted from the genomic information of four *Shigella* species (*S. dysenteriae*, *S. flexneri*, *S. boydii*, and *S. sonnei*) (http:www.p2cs.org/, accessed on 1 May 2022) [48] were compared with those of 29 HKs and 32 RRs of *E. coli* K-12 (Table 1). As a result, they were found to be significantly conserved across the four *Shigella* species and *E. coli* K-12 (Table 1). In particular, five TCSs, CpxA/CpxR, EnvZ/OmpR, EvgS/EvgA, ArcB/ArcA, and PhoQ/PhoP, are known to be involved in virulence (Table 2, Figure 2). These TCSs are described in detail in the following sections.

## 5. EnvZ/OmpR and the Invasion of Epithelial Cells in Response to Osmolarity

One of the first TCSs shown to be involved in the regulation of the invasiveness of *Shigella* is the EnvZ /OmpR system, which controls the expression of outer membrane porins in response to changes in osmolarity in the extracellular environment [52,58]. EnvZ is an inner membrane sensor with kinase activity, and OmpR is the cognate cytoplasmatic regulator acting as a DNA binding protein [59,60]. At high osmolarity, the cytoplasmatic domain of EnvZ increases its auto-phosphorylation at His-243. The phosphoryl group is then transferred to OmpR at a conserved aspartic acid residue (Asp55) [61]. In the phosphorylated state, OmpR controls the expression of two major outer membrane proteins, OmpC and OmpF. It is well known that *E. coli* responds to changes in the environmental osmotic strength by changing the ratio of the OmpC and OmpF porins in the outer membrane [58]. OmpC, which produces slightly smaller pores, predominates at high osmolarity, whereas OmpF predominates at lower osmolarity. This strategy envisages that the smaller pores of OmpC could hamper the entry of large molecules when bacteria encounter high osmolarity conditions, as inside the host. On the contrary, in a low osmotic external environment, the larger pores of OmpF could favor the diffusion of nutrients at lower concentrations. According to this model, it has been shown that in *E. coli* at high osmolarity, the binding of phosphorylated OmpR at the *ompC* promoter activates the transcription of *ompC*, which provokes the formation of a repression loop at the *ompF* promoter that occludes RNA polymerase binding and subsequent *ompF* transcription [61]. Using a fusion of the *lacZ* gene with one of the *Shigella* pINV *vir* genes (*mxiC*), it has been shown that moving from low to high osmolarity induces a significant increase in β-galactosidase activity [52]. The high expression of *vir* genes under osmotic pressure correlates well with the osmotic conditions found by *Shigella* in the intra- and extracellular compartments of human hosts. Studies performed with mutants of the *ompB* locus, containing the *ompR-envZ* operon, demonstrate that the presence of a mutation in the *envZ* gene (*envZ*::Tn10) reduces the intracellular survival, impairs the ability to form plaques on confluent lawns of HeLa cells, and induces delayed and attenuated keratoconjunctivitis in the Sereny test (Table 2). Moreover, this mutation leads to the decreased β-galactosidase activity of *vir-lacZ* fusion in both low and high osmolarity conditions [52]. The effect on *Shigella* virulence is more severe in cases of deletions covering the entire operon (Δ*ompB*). The lack of the *ompB* operon hampers the colonization of epithelial tissue cultures and abolishes the expression of the *vir-lacZ* fusion in both low and high-osmolarity conditions. Since the *ompB* mutant does not produce OmpC and OmpF porins, it has been investigated whether the loss of one or both porins could affect the ability of *S. flexneri* to survive and grow in the host cells [53]. While the *ompF* mutant behaves like the *S. flexneri* wt strain in all in vitro and in vivo assays, the Δ*ompC* mutant exhibits a severely altered virulence phenotype. In particular, similarly to the Δ*ompB* mutant, the ability to spread intercellularly and kill the host cells is impaired. As a result, *S. flexneri* strains carrying Δ*ompB* and Δ*ompC* deletions remain trapped within the initially invaded cells, losing the capacity to form extracellular protrusions and to move towards adjacent cells. Furthermore, the lack of OmpC renders *S. flexneri* unable to elicit keratoconjunctivitis in Sereny tests, as previously observed in the absence of the *ompB* locus, while the invasion and killing of macrophages remain unaffected. Ectopic expression of OmpC porin in a Δ*ompB* background restores the virulence phenotype, indicating that the effect of OmpR and EnvZ on virulence is exerted mainly through the OmpC protein [53]. 

While the EnvZ/OmpR system is typically associated with the regulation of OmpC and OmpF porins in response to osmolarity, transcriptomics and chromatin immunoprecipitation analyses show that in *E. coli*, it acts as a global regulatory system [62]. Indeed, silencing of EnvZ and OmpR causes a severe change in the global expression profile, including increased expression of genes encoding multidrug resistance (MDR) efflux pumps and systems involved in the synthesis and transport of siderophores. Moreover, it controls the activation of ferric uptake genes, thus contributing to cell iron homeostasis [63]. In *E. coli* and *S. Typhimurium*, OmpR has been shown to modulate gene expression in response to acid stress [64]. Altogether these data lead to the hypothesis that in *Shigella*, the EnvZ/OmpR system may contribute to the full expression of the virulence phenotype by controlling other determinants involved in intracellular fitness. 

## 6. CpxA/CpxR and the Regulation of *vir* Genes in Response to pH 

In addition to temperature and osmolarity, changes in environmental pH also modulate virulence gene expression in *Shigella*. In particular, at low pH, the *Shigella* invasive phenotype is repressed by approximately 10-fold compared to a pH of 7.4. pH-dependent regulation requires TCS CpxA/CpxR [49]. This system is known to respond to alkaline pH. In addition, it also senses a large variety of stimuli typically associated with cell envelope stress and is able to give rise to misfolded envelope proteins [65]. When the membrane-anchored sensor kinase CpxA is activated, it promotes its autophosphorylation at the conserved histidine residue His 248 and then phosphorylates the CpxR protein. In *E. coli*, the phosphorylated CpxR (CpxR-P) acts as a transcription factor able to activate a large number of genes. In particular, CpxR-P positively regulates several genes encoding periplasmatic proteases and chaperones such as the heat shock protease DegP, the peptidyl prolyl *cis/trans* isomerases PpiA and PpiD, and the disulfide oxidoreductase DsbA [65]. These proteins are able to degrade or fold the misfolded proteins, thereby mitigating envelope stress. In addition, CpxR-P can repress several genes, including those required for motility and chemotaxis. The Cpx system includes a third protein, CpxP, located in the periplasmatic space [66] and encoded by a gene directly activated by CpxR-P. This protein interacts with the CpxA sensory domain, downregulating CpxA kinase activity. The current model predicts that the Cpx response is triggered when the CpxP protein is titrated by misfolded envelope proteins [67]. Recently, it was reported that the NplE lipoprotein also directly interacts with the CpxA sensor through its N- terminal domain and activates the Cpx system when lipoprotein trafficking is altered [68].

Genetic and molecular approaches have shown that the *Shigella* invasive phenotype, in response to changing pH, is linked to the expression of the *virF* gene, which encodes the major regulator of the pINV virulence genes [49]. Indeed, *virF* transcription is strongly reduced at low pH. This also holds true for the β-galactosidase activity of *virF-lacZ* transcriptional fusion. Further experiments demonstrate that the repression of *virF* at low pH is only partially alleviated in the absence of the nucleoid-associated protein H-NS and that a mutation of the *cpxA* gene restores a high level of VirF even under acid conditions. This *cpxA* mutation gives rise to a shorter CpxA protein, able to constitutively activate the Cpx pathway. On the other hand, knockout of the cognate regulator *cpxR* completely abolishes the expression of *virF* at both low and high pH, indicating that the CpxR protein is required for *virF* expression. Indeed, CpxR binds directly to the *virF* promoter by recognizing a region spanning from −37 to −103 nucleotides from the transcription start site [50]. The binding capacity of CpxR is enhanced when the protein is in the phosphorylated form. This evidence has led to speculation that at low pH, CpxA might act as a phosphatase on the phosphorylated CpxR, thus preventing it from binding to the *virF* promoter [50]. Further studies, based on a screening of *S. sonnei* mutants having a low expression level of the T3SS system, have led to the identification of two more mutants carrying a defective *cpxA* gene [51]. The *Shigella* T3SS system is encoded by a large operon located on the pINV virulence plasmid and is positively controlled by the VirB regulator, which in turn requires VirF for its activation. A detailed analysis of these *cpxA* mutants indicates that under neutral pH conditions, CpxA is required for post-transcriptional processing of VirB. The molecular mechanisms responsible for the post-transcriptional control of VirB are not yet clearly elucidated. However, even in the case of TraJ, a regulatory protein required for F-mediated conjugation, CpxA-mediated post-transcriptional regulation has been observed [69]. Taken together, these observations indicate that the Cpx system may act at both transcriptional and post-transcriptional levels to modulate the full expression of the *Shigella* virulence phenotype (Table 2). 

Many studies have highlighted how the Cpx system is involved in controlling the infection process in several bacterial pathogens [70]. As previously reported for the *virF* gene, the Cpx response can modulate the transcription level of key regulators of virulence genes. This is the case, for example, in the *hilA* gene of *Salmonella typhimurium*, which encodes the master regulator of invasion genes and whose expression is greatly reduced in the absence of a functional CpxA protein [71]. In uropathogenic *E. coli*, binding of the phosphorylated CpxR to the switch region that controls the phase-variable expression of *pap* operons hinders the expression of P pili [72]. Moreover, in *Legionella pneumophila* CpxR plays a relevant role in virulence by controlling the transcription of numerous components of the Dot/lcm T4SS and its effectors required for the interaction with the host cell [73]. The relevance of the Cpx system is further underscored by its role during the early stages of the infection process in several pathogens. Indeed, the Cpx response is involved in the expression of surface structures which are responsible for the adhesion of bacteria to host cells. A well-studied example of an adhesion organelle whose assembly is controlled at the post-transcriptional level by the Cpx system is the type IV bundle-forming pilus (BFP) of enteropathogenic *E. coli* (EPEC) [74]. Considering that in many bacteria, the number of genes regulated by the Cpx system is continuously expanding [65,75], it is possible to speculate that the Cpx response may modulate many other genes involved in the adaptation of *Shigella* to the host, thus further contributing to bacterial fitness during intracellular life.

## 7. ArcB/AcrA and the Activation of Iron Transport Systems

The success of the *Shigella* invasive process is strictly dependent on the presence of specific environmental conditions, such as oxygen availability, which is advantageous for the pathogen [76]. *S. flexneri* typically infects the colon of higher primates, which implies that this pathogen must cope with a limited oxygen environment in order to deploy its full virulence potential. The aerobic respiration control (Arc) TCS is a complex, well-conserved and ubiquitous signal transduction system that senses oxygen availability or consumption and regulates energy metabolism [77,78]. The presence of the ArcB/AcrA TCS in *S. flexneri* as well as in many other pathogenic and not pathogenic facultative anaerobic bacteria, allows them to switch to fermentation or anaerobic respiration in order to optimize energy production when aerobic respiration is not possible [79]. ArcB/AcrA is a canonical TCS activated via a phosphoryl relay and consisting of a membrane-associated sensor kinase, ArcB, that autophosphorylates upon detection of oxygen consumption and then phosphorylates the cytosolic response regulator, ArcA [80,81] (Figure 1). The activation of the ArcA transcription factor mainly results in the suppression of aerobic metabolic pathways [78]. The ArcB/AcrA TCS has been extensively studied, mainly in *E. coli*, due to its key role in bacterial metabolism and its pleiotropic effect on cellular processes [78,81]. Indeed, the importance of ArcB/AcrA-mediated regulation becomes crucial not only when bacteria have to switch on the expression of anaerobic respiration genes when oxygen is poorly available but also to balance the activity of oxygen-dependent and oxygen-independent proteins. This is what usually happens during infection when pathogens need to adapt rapidly to and survive in the host environment. For this reason, ArcA is considered a global regulator that directly or indirectly affects the expression of more than 1100 genes [79,82]. Indeed, the activity of the Arc system is strictly related to different metabolic pathways working together with other transcriptional regulators such as fumarate and nitrate reductase regulator (FNR) [83]. ArcA and Fnr share common features, such as being responsive to anoxic conditions [83]. In *S. flexneri*, ArcA and Fnr are important for intracellular growth and cell-to-cell dissemination. Indeed, infection of Henle cells with an *arcA fnr* double mutant performed under anaerobic conditions does not give rise to plaques on a confluent cell lawn. This inhibitory effect is not fully displayed when the infection is carried out using *S. flexneri* harboring only a single mutation (*acrA* or *fnr*), suggesting that the functions of the AcrA and Fnr regulators partially overlap [54]. 

How does the ArcB/AcrA TCS impact the infection process of *S. flexneri*? In *S. flexneri*, oxygen availability, sensed by the action of the Arc system, is a key signal for the regulation of iron acquisition [54]. As observed in other pathogenic bacteria, also in *Shigella*, different iron transport systems exist. They allow the cell to acquire both ferrous (Fe^2+^) and ferric iron (Fe^3+^), either free or complexed with different carriers, in all the different environments the bacterium encounters within the human host [84]. *Shigella* can acquire ferrous iron, a form which is abundant in anaerobic environments, through Sit and Feo transporters and can also uptake the less soluble ferric iron through specific transporters that use siderophores with high affinity for this iron form [84]. In particular, to uptake ferric iron, the predominant form under aerobic conditions at neutral pH, *S. flexneri* produces aerobactin (Iuc), a hydroxamate siderophore synthesized by the *iucABCD* operon and transported by the Iut transport system [54,85]. In addition, through the Fhu transport system, *Shigella* is able to transport the fungal siderophore ferrichrome [54]. The expression of the uptake, transport, and storage systems is highly regulated in response to environmental conditions, ensuring that the pathogen receives a continuous supply of iron, essential for survival and infection, without incurring iron toxicity side effects [86]. The regulation of the iron acquisition genes in response to oxygen availability and their impact on *Shigella* virulence were specifically addressed by Boulette and Payne [54]. In particular, microarray analyses revealed that the transcription of *feo* genes is induced under anaerobic conditions, while *sit* and *iuc* gene expression increases when oxygen is abundant. This aerobic or anaerobic gene expression is mainly driven by two regulators: ArcA, which directly acts as a repressor of *iuc* genes under anaerobic conditions and indirectly induces aerobic expression of *sit* genes, and Fnr, which in conjunction with ArcA, induces *feo* transcription. Interestingly, this tight regulation has a relevant effect on the infection process of *Shigella*. Plaque assays performed under anaerobic conditions show that Iuc, Sit, and Feo are the main transport systems used by *S. flexneri* to acquire iron during infection of cultured cells. Indeed, the lack of the three genetic loci impairs plaque formation, both aerobically and anaerobically, in Henle cell monolayers (Table 2). Moreover, a detailed analysis of the contribution of each iron transport system reveals that, in line with the results obtained by transcriptional analysis, the double mutant *sit iuc* is impaired in plaque formation aerobically since *feo* expression is regulated by ArcA under anaerobic conditions. Similar results are obtained with a *feo sit* double mutant whose ability to form plaques is drastically compromised under anaerobic conditions due to ArcA repression of the aerobactin system [54]. The influence of oxygen through ArcA regulation is not only visible in iron transport systems whose promoters, in most cases, host an ArcA binding site (i.e., *iuc* and *feo* promoters) but also affects iron uptake by directly binding the *fur* promoter and repressing its transcription [54]. Fur, the Ferric Uptake Regulator, is the main regulator of iron uptake genes and acts as a transcriptional repressor in response to iron availability. In particular, under high iron levels, Fur binds Fe^2+^ and represses transcription of the genes encoding siderophore and ferrous iron transporters. The activity of Fur is thus enabled by the absence of ArcA repressor, and its importance for *Shigella* virulence is demonstrated by the reduced or absent formation of plaques when Fur is deregulated (overexpression or lack, respectively) [54]. In essence, the expression of appropriate iron acquisition systems, tightly regulated by ArcB/AcrA TCS under the aerobic or anaerobic conditions typically encountered by *Shigella* during the invasive process, represents an important weapon that the bacterium can deploy for successful infection.

## 8. The Role of PhoQ/PhoP during Later Stage of *Shigella* Infection

PhoQ/PhoP is a TCS constituted by histidine kinase PhoQ, localized in the inner membrane and acting as a sensor. PhoP is a cytoplasmic regulator that can bind to specific DNA sequences and regulate their transcription. In response to a wide range of environmental stimuli, PhoQ autophosphorylates using available ATP, and then the phosphate group is transferred to a conserved aspartate residue of the N-terminal domain of PhoP. The phosphorylated PhoP can bind a conserved DNA sequence, called PhoP-box, and induce or repress the expression of selected genes [87]. Among the signals that can be sensed by PhoQ, the concentration of periplasmic Mg^2+^ (and other divalent cations such as Ca^2+^ and Mn^2+^) was the first to be identified [87]. The expression of PhoP-activated genes is repressed by high concentrations of magnesium ions (millimolar), whereas low concentrations (micromolar) of the same ion induce the phosphorylation of PhoP [88,89,90]. Other inducing signals include, but are not limited to, mildly acidic pH even at millimolar concentrations of divalent cations [91], cationic antimicrobial peptides such as C18G and LL-37 [92], hyperosmotic stress caused by NaCl concentrations higher than 150 mM [93], and disruption of the oxidizing environment of the periplasm [94]. Additionally, repressing signals, such as long-chain unsaturated fatty acids, have been described [95]. Some signalling molecules compete with each other to bind to the PhoQ [92]. 

In experimental studies performed both on *Salmonella enterica* and *Escherichia coli*, the PhoQ/PhoP regulons have been reconstructed, and, interestingly, their overlap is small, even if the regulatory system is extremely conserved. PhoQ/PhoP-regulated *Salmonella* genes are mainly involved in central intermediary metabolism, synthesis, and modification of the cell envelope, cation transport, and drug sensitivity. Most *E. coli* genes that are directly regulated by PhoQ/PhoP are involved in general metabolism and cell and membrane structure [56]. The PhoQ/PhoP system has been extensively studied in *S. enterica*; early investigations concerning its function were performed through *Tn10* insertion mutants that were not able to survive inside mouse macrophages, suggesting a role as an important virulence determinant involved in response to various environmental stimuli [96]. During the infection, *Salmonella* enters the macrophages and is able to survive within the phagosome, where many of the inducing signals mentioned above are present [97]. 

The importance of the PhoQ/PhoP system in the virulence of *Shigella* was recognized later [98], and to date, the available data are still limited. The first evidence of the involvement of the PhoQ/PhoP system in the virulence of *S. flexneri* stems from observations on the ability of this pathogen to survive inside the cationic, antimicrobial molecules-rich phagocytic vacuole of polymorphonuclear leucocytes (PMNs) [98]. In the same study, Moss and colleagues [98] employed a murine pulmonary infection model to analyze PhoP-mediated inflammation in the mouse lung. They observed that even if the colonization efficiency is similar to the wild-type *Shigella* strain, *phoP* mutants were cleared more rapidly, and the inflammatory response they were able to induce persisted for a shorter time. Furthermore, they noticed that the *phoP* mutants could be killed faster by PMNs and were more susceptible to the antimicrobial activity of peptides such as magainin-2 (Table 2). In the wild-type strain, a similar phenotype is the consequence of exposure to a high concentration (25 mM) of Mg^2+^, an environmental condition known to cause the dephosphorylation of the cytoplasmic regulator PhoP [98]. The PhoQ/PhoP system is involved in the regulation of the expression of genes essential for resistance to cationic peptides, such as polymyxin B and colistin and it has been shown that at least 19 single amino acid substitutions can increase resistance to polymyxins in several Gram negative bacteria [99]. More recently, studies have been performed using a *phoPQ* deletion mutant to prove both in vitro and in vivo that PhoQ/PhoP is significantly involved in the regulation of the virulence of *Shigella*. In HeLa and CaCo-2 cells, the *ΔphoPQ* deletion mutant shows a considerable reduction of invasion rate and cannot induce membrane ruffling. The latter effect is commonly observed in HeLa cells following the cytoskeleton changes triggered by *Shigella* virulence factors and is crucial for efficient spreading to adjacent epithelial cells [57]. Among those virulence factors, IcsA is required to induce the polymerization of actin comet tails utilized by *Shigella* for intra- and intercellular spreading [21]. In the *ΔphoPQ* strain, *icsA* is among the genes differentially expressed at a middle-log phase in the LB medium [57]. The PhoPQ-mediated regulation of *icsA* expression has been validated through various assays by Lin and coworkers [57]. In the promoter region of the *icsA* gene, they found a conserved PhoP-box motif, and they also observed a marked reduction in the transcriptional levels of *icsA* in the *phoPQ* deletion mutant. Moreover, the authors showed that the decreased invasion rate and membrane ruffles phenotype of the *ΔphoPQ* strain could be rescued by ectopic expression of the *icsA* gene. Those investigations have been corroborated by in vivo experiments based on the guinea pig Sereny Test [57]. In addition to *icsA*, many other genes have been included in the *Shigella* PhoQ/PhoP regulon. Among them, there is the *shf-wabB-virK-msbB2* operon encoded on pINV, the large virulence plasmid shared by the four *Shigella* species [57]. The genes *msbB2* and *msbB1* are paralogous. The latter is harbored on the bacterial chromosome, and the corresponding proteins (MsbB2 e MsbB1) share 69% homology. Both proteins are involved in the last stages of lipid A synthesis: they catalyze the transfer of the myristate residue from myristoyl-ACP (Acyl Carrier Protein) to penta-acyl lipid A, resulting in a fully exa-acylated final product. The expression of *msbB2* is enhanced by magnesium-limiting conditions and is dependent on the activity of the PhoQ/PhoP system [100]. Interestingly, *msbB1* does not require PhoQ/PhoP for its expression, which is also not affected by the availability of magnesium in the medium. Such a peculiar condition may allow *Shigella* to produce the required amount of MsbB proteins in different environments, allowing the pathogen to increase its possibility of surviving during host infection [100]. 

## 9. Role of EvgS/EvgA during Intracellular Life of *Shigella*

The TCSs described so far in this review (EnvZ/OmpR, CpxA/CpxR, ArcB/A, and PhoQ/PhoP) are conserved in various Gram-negative bacteria, respond to similar signals, and exhibit similar functional outputs. However, some TCSs are conserved only in specific species. The EvgS/EvgA system is such a case. This system is highly conserved in *E. coli* and *Shigella* but is not found in other bacteria [101,102]. The virulence-related BvgS/BvgA system in *Bordetella pertussis* (the etiological agent of whooping cough) [103] and the KvgS/KvgA system in *Klebsiella pneumoniae* [104] are similar to EvgS/EvgA. To our knowledge, no further examples of similar systems have been reported to date.

The EvgS/EvgA system was first identified in the genome of *E. coli* strain K-12. Its naming refers to its similarity to the BvgS/BvgA system in *B. pertussis* [105]. EvgS is a hybrid HK that serves as a sensor, and EvgA is its cognate RR belonging to the NarL family (Figure 1, Table 1). The genes encoding these proteins are part of the *evgAS* operon, which is adjacent to the *emrKY* operon and encodes the EmrKY efflux pump, a member of the Major Facilitator Superfamily (MFS) of efflux pumps [106]. EmrKY functions as a tripartite efflux pump, with TolC as an outer membrane factor. As shown by Kato et al. (2000) [106], EvgA binds to the promoter region of the *emrKY* operon and directly regulates its expression. The signals to which the EvgS sensor responds were originally not known, but the authors found a constitutively active mutant (EvgS1; with an amino acid substitution F577S) and used it to confirm that the activation of EvgS induced the expression of EmrKY and conferred drug resistance to *E. coli* cells. Moreover, ectopic induction of the EvgA regulator was shown to promote the expression of an additional efflux pump, MdtEF, encoded by the *gadE-mdtEF* operon [107], which belongs to the Resistance Nodulation Division (RND) family of efflux pumps and functions with TolC as outer membrane channel. Transcriptome analyses using the constitutively active EvgS1 mutant revealed that the activation of the EvgS sensor induced the expression of several efflux pump and efflux pump-related genes, such as *emrKY, mdtEF, acrA, mdfA*, and *tolC* [108]. Among these genes, *emrKY*, *mdtEF*, and *tolC* mainly contribute to the drug resistance conferred by the EvgS/EvgA system [109]. The concept that TCSs regulate the expression of efflux pumps suggests that multidrug resistance of pathogens can be induced. 

In addition to efflux pump genes, there are many other genes whose expression is induced by EvgS activation. By transcriptome analysis of EvgS/EvgA involving activation of the system via EvgA overexpression, Masuda and Church focused on the elevated expression of genes related to acid resistance [110,111]. EvgS activation or overexpression of EvgA confers acid resistance to exponentially growing *E. coli* cells by switching on a transcriptional cascade of regulators, EvgA, YdeO, and GadE, which induce the expression of various acid resistance (AR) genes, such as *gadA*, *gadB*, *gadC*, *hdeAB*, *hdeD*, and *ydeP* [111,112]. EvgS/EvgA induces the expression of AR2 genes (*gadA*, *gadB*, and *gadC*), which confer tolerance to extreme acidic conditions, i.e., between pH 2–3. The EvgS/EvgA system is also reported to be activated by mild acidic pH [113] and has been recognized as a TCS that is related to acid resistance rather than drug resistance. Further studies of the EvgS sensor reveal that a number of coordinated conditions are required to activate EvgS. *E. coli* needs to be grown in a mildly acidic (pH 5–6) minimal medium, with high concentrations (≥100 mM) of potassium or sodium ions in an aerobic environment [38,114]. Furthermore, EvgS activity is repressed in the presence of the bacterial signalling molecule indole [41]. This suggests that EvgS activity is repressed when *E. coli* cells reside in the colon of their mammalian host, where the environment is anaerobic, and the cells may encounter indole. The strong AR that EvgS/EvgA confers to *E. coli* cells can be connected to their tolerance to the harsh acidic environment of the stomach of their mammalian host. 

Does *E. coli* experience specific EvgS sensor-activating conditions (i.e., mildly acidic, low nutrient content, high K^+^ or Na^+^ concentration, and aerobic environment) before entering the stomach? While this is an unresolved question, speculation on the function that the highly conserved EvgS/EvgA system might play in *E. coli* can be put forward. Considering that a growing body of work indicates that multidrug resistance (MDR) efflux pumps are involved not only in antibiotic resistance but also in virulence, the role of TCSs becomes even more relevant in the physiology of pathogenic bacteria and in their interactions with host cells [115,116,117]. In this context, a recent study of MDR efflux pumps in *S. flexneri* reveals that among its 14 efflux pumps (a pool which has been retained by *S. flexneri* out of the 20 efflux pumps characterized in *E. coli*), the expression of *emrKY* is highly induced when *S. flexneri* strain M90T infects U937 macrophage-like cells [55]. This induction is EvgA-dependent, indicating that EvgS/EvgA becomes active when *S. flexneri* invades macrophages. Following *S. flexneri* infection, the intracellular pH of macrophages is reduced slightly, i.e., to approximately pH 6. The mild acidic pH, high K^+^ concentration, and oxidative/low-nutrient environment of the macrophage cytosol coincide with the signals known to activate *E. coli* EvgS. After *S. flexneri* invades macrophages, EvgS/EvgA is rapidly activated and induces the expression of *emrKY*. In the same study [55], the expression of efflux pumps in *S. flexneri* was examined in *S. flexneri*-infected Caco-2 epithelial cells. As opposed to macrophage-like cells, the level of *emrK* expression inside Caco-2 cells decreases, indicating that EvgS/EvgA of *S. flexneri* is activated inside macrophages and not in epithelial cells, presumably because of the neutral cytosolic pH of the latter. In essence, the activation of signal transduction pathways through EvgS/EvgA once *S. flexneri* invades macrophages contributes to the fitness of *Shigella* in this cellular environment because deleting *emrKY*, directly regulated by EvgS/EvgA, results in an intracellular growth disadvantage of *Shigella* [55] (Table 2, Figure 2). Furthermore, *E. coli* cells with mutations involving *emrK* and *emrY* are sensitive to mitomycin C, UV irradiation, and H_2_O_2_, indicating that EmrKY may eliminate toxic metabolites induced by DNA damage [118]. These findings may provide further clues for understanding how EmrKY functions to support the survival of *S. flexneri* in macrophages. 

As mentioned above, EvgS/EvgA also induces the expression of the MdtEF efflux pump in *E. coli* K-12 [107,119]. Interestingly, while in *S. flexneri* the *gadE-mdtEF* operon is disrupted, it is conserved in the other three *Shigella* species, *S. boydii*, *S. sonnei*, and *S. dysenteriae* (Y. Eguchi, personal communication). Further studies will be necessary to clarify whether EvgS/EvgA will induce the expression of MdtEF in *Shigella* species to enforce intracellular survival. Recently, using Adherent Invasive *E. coli* (AIEC) strain LF82 as a model, it was reported that EvgS/EvgA and PhoQ/PhoP TCS contribute to the intracellular survival of AIEC strains in macrophages [120] and that the MdtEF efflux pump is induced in macrophages and increases the bacterial fitness in this harsh environment [121]. 

## 10. Molecular Mechanism of TCS Network

As previously outlined, five TCSs (CpxA/CpxR, EnvZ/OmpR, EvgS/EvgA, ArcB/ArcA, and PhoQ/PhoP) are known to control virulence in *Shigella* (Table 2, Figure 2). Multiple TCSs are involved in controlling virulence and also in other pathogenic bacteria [1,2]. For example, 5 TCSs are found in *S. typhimurium* (PhoQ/PhoP, PmrB/PmrA, EnvZ/OmpR, SsrA/SsrB, and PmrB/PmrA) [1], as many as 15 are found in *Pseudomonas aeruginosa* [122], 3 are found in *Staphylococcus aureus* (AgrC/AgrA, SaeS/SaeR, and SrrA/SrrB) [2], and 5 are found in pathogenic *Escherichia coli* (CpxA/CpxR, EnvZ/OmpR, EvgS/EvgA, RcsC/RcsD/RcsB, and QseC/QseB) [33]. These TCSs contribute to invading host cells, evading the immune system, adapting to the intracellular environment, and proliferating inside host cells [1,122]. To control the expression of virulence factors at different stages during infection, sophisticated interactions among several TCSs, which form a TCS network, may play important roles [1,122]. In this section, considering the significant overlap between *Shigella* and *E. coli* TCSs (Table 1), the molecular mechanisms of the well-studied network of *E. coli* TCSs are introduced. 

The virulence-related TCSs previously addressed have been reported to be connected by small HK-modulating proteins in *E. coli* [123]. The CpxA/CpxR and EnvZ/OmpR systems are linked by the MzrA protein, which is a small protein of 127 amino acid residues that resides in the inner membrane. It directly binds to EnvZ via interactions in the periplasm and activates the EnvZ/OmpR system [124]. The residues D51 and I78 of MzrA and the VVPPA poly-proline motif in the EnvZ periplasmic domain have been reported to be involved in interactions between the two proteins [124,125]. MzrA is encoded by the *yqjA-mzrA* operon. There are σ^70^(p1)- and σ^E^(p2)-dependent promoters upstream of the *yqjA-mzrA* operon, and the expression of this operon is positively regulated by the CpxA/CpxR system [126]. Thus, CpxA/CpxR is functionally coupled to EnvZ/OmpR via MzrA. The expression of *mzrA* is increased significantly by the activation of the sensor CpxA. However, the level of *mzrA* expression only decreases modestly in the absence of CpxR. This has been explained by the involvement of additional regulators, such as the alternative sigma factor σ^E^, in the regulation of *mzrA* [127]. The MzrA proteins of *E. coli* MG1655 and *S. flexneri* M90T display 97.6% amino acid sequence identity, and in the case of EnvZ, 100% identity is observed. The D51 and I78 residues in MzrA are present in *S. flexneri*, and the CpxA/CpxR system is assumed to be associated with the EnvZ/OmpR system in *S. flexneri*. Considering that CpxA/CpxR and EnvZ/OmpR are both involved in the invasion of host cells (Table 2), it is possible to hypothesize that the link between these two TCSs may influence the virulence of this pathogen.

The activation of the EvgS/EvgA system also induces the expression of a small (65 residues) protein, SafA. SafA is encoded by the *safA-ydeO* operon, and EvgA directly binds to the promoter region of this operon to induce its expression [112]. SafA resides in the inner membrane and spans the membrane once, with its C-terminal region facing the periplasm. The C-terminal region of SafA binds directly to the periplasmic sensor domain of PhoQ and activates PhoQ by enhancing its autophosphorylation activity [128,129]. Thus, EvgS/EvgA is connected to PhoQ/PhoP via SafA. Although MzrA is a relatively well-conserved protein of the *Enterobacteriaceae* family [127], SafA is only found in *E. coli* and *Shigella*, which is also the case for EvgS/EvgA, as reported above. This suggests the co-evolution of these two proteins. Between *E. coli* MG1655 and *S. flexneri* M90T, SafA and PhoQ show 98.5% and 99.8% amino acid sequence identity. This implies that EvgS/EvgA and PhoQ/PhoP may also be connected in *Shigella*. Both EvgS/EvgA and PhoQ/PhoP contribute to intracellular survival in host cells and linking these two TCSs may enhance their ability to survive intracellularly.

The PhoQ/PhoP system also induces the expression of IraM, a small cytoplasmic protein of 107 residues [130]. In *E. coli*, IraM binds to another RR, namely RssB. RssB binds to the stress response σ factor RpoS and escorts RpoS to ClpXP for degradation. Thus, the binding of IraM to RssB prevents RpoS degradation [130]. When EvgS/EvgA is activated in exponentially growing *E. coli* cells, PhoQ/PhoP is activated via SafA and RssB is trapped by IraM, resulting in the accumulation of RpoS. However, the amino acid sequence of IraM of *S. flexneri* M90T is only 61.7% identical to that of *E. coli* MG1655, and the PhoQ/PhoP-IraM-RssB-RpoS line may not be functional in *S. flexneri*.

The presence of complicated TCS networks has been reviewed in other pathogens, such as *S. typhimurium* and *P. aeruginosa* [122,123]. Understanding the complex relationships between TCS systems during infectious processes of *Shigella* is an interesting challenge that may also open new perspectives in controlling the expression of virulence genes and intracellular survival. 

## 11. Prospects: Drug Discovery Targeting TCS Network

TCSs are emerging as crucial regulators of the virulence phenotype in an increasing number of life-threatening bacterial pathogens. Recently, it has been shown that in a *Staphylococcus aureus* mutant lacking the genes encoding 15 nonessential TCSs, including HK and RR, virulence is highly attenuated during infections in mouse models [131]. In *Salmonella enterica* serovar Typhi, Murret-Labarthe et al. [132] individually deleted the genes encoding each of the 30 RRs. Deletion of most RR-encoding genes (24 of 30) resulted in significant changes during interaction with host cells (epithelial cells and macrophages) [132]. This suggests that multiple TCS signaling systems contribute significantly to the expression of virulence factors and the adaptation and survival of the pathogen in animal hosts. It is, therefore, reasonable to assume that an HK inhibitor targeting multiple HKs could represent an innovative antivirulence drug against pathogenic bacteria [133]. Indeed, to identify molecules with broad anti-HK activity, Wilke et al. [134] targeted the ATP-binding domain (CA) of HKs and performed high-throughput screening of 53,000 diverse small molecules, but the most potent compounds proved cytotoxic [134,135]. Other studies led to the identification of an HK inhibitor, waldiomycin, able to bind to the Hbox region commonly conserved in the DHp domain of HKs [136]. This compound inhibits Class I HKs (Figure 1, Table 1), including EnvZ, PhoQ, CpxA, EvgS, and ArcB, which are involved in the regulation of virulence in *Shigella* (Table 2, Figure 2), but not a class II HK (CheA, Table 1). Thus, in general, Hbox inhibitors can be expected to be promising therapeutic agents to control the virulence of pathogens, including *Shigella*. 

The role of TCSs in *Shigella* pathogenicity is still underestimated. In particular, only global approaches will allow us to understand which and how many genes are modulated by the loss of function of one or more TCSs. In addition to the TCSs presented in this review, several others found in *E. coli* are conserved in *Shigella*, but their role in bacterial interactions with the host has not yet been elucidated. Overall, understanding the mechanisms of TCS-mediated virulence regulation in a pathogen such as *Shigella*, which is extremely well adapted to its human host, constitutes a fascinating and complex challenge that may also open interesting prospects for novel antibacterial therapies. 

## Figures and Tables

**Figure 1 biomolecules-12-01321-f001:**
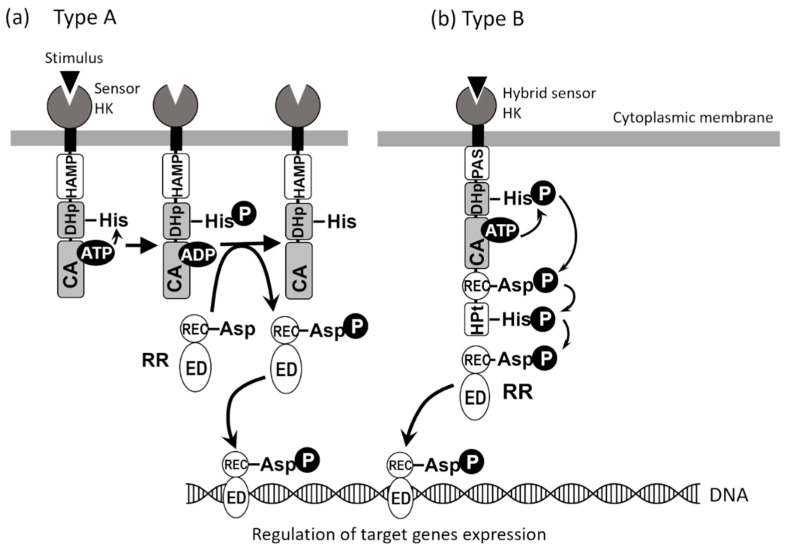
**Schematic representation of the two types of TCSs.** (**a**) In Type A, the sensor domain at the N-terminal region of HK is followed by HAMP (a domain found in Histidine kinases, Adenyl cyclases, Methyl-accepting proteins, and Phosphatases), DHp (Dimerization and Histidine phosphotransfer domain), and CA (Catalytic and ATP-binding domain) at the C-terminal region. (**b**) In Type B, the sensor domain at the N-terminal region of the hybrid sensor is followed by PAS (Per-Arnt-Sim domain), DHp, CA, REC (Receiver domain), and HPt (Histidine Phosphotransfer domain) at the C-terminal region. RR consists of the REC domain (N-terminal) and ED (Effector domain) (C-terminal).

**Figure 2 biomolecules-12-01321-f002:**
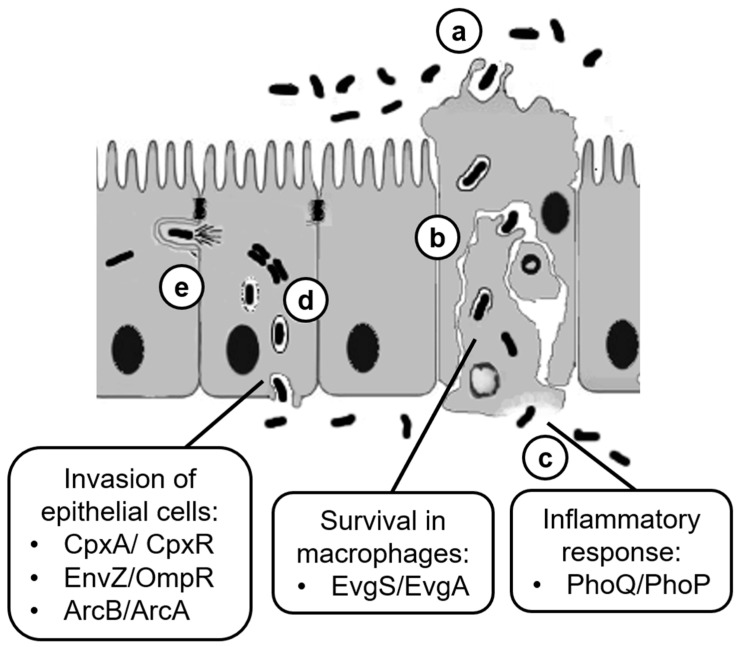
**Role of TCS systems in the different steps of the invasive process of Shigella.** (a) *Shigella* crosses the colonic epithelium via M cell translocation. (b) *Shigella* invades the resident macrophages, induces lysis of the phagolysosome, surviving the cell attack, and (c) escapes the macrophages by triggering inflammatory cell death (pyroptosis) with a consequent release of proinflammatory cytokines. (d) *Shigella* invades epithelial cells of colonic mucosa by stimulating macropinocytosis, induces vacuole lysis, multiplies in the cell cytosol, and (e) spreads to adjacent cells using actin-based motility. The involvement of TCS systems in the various steps is reported in the boxes.

**Table 1 biomolecules-12-01321-t001:** Distribution of TCSs in *Shigella* spp.

*Escherichia coli* K-12 MG1655				*S.dysenteriae* 1617	*S.flexneri* 2a str.301	*S.boydii* Sb227	*S.sonnei* 53G
HK_29_ **	RR_32_	Stimulus	Ref	HK_23_/RR_29_	HK_25_/RR_30_	HK_26_/RR_30_	HK_30_/RR_32_
**OmpR family**							
PhoR (a)	PhoB	Phosphate	[35]	HK/RR *	HK/RR	HK/RR	HK/RR
CusS (a)	CusR	Cu^2+^	[36]	np	HK/RR	np	HK/RR
KdpD (a)	KdpE	PtsN. K^+^, ATP, Ionic strength	[36,37]	RR *	np *	HK/RR	HK/RR
TorS (b)	TorR	TorT, TMAO	[37]	np *	RR *	HK/RR	HK/RR
PhoQ (a)	PhoP	Ni^2+^, Mg^2+^, Ca^2+^, SafA Antimicrobial peptide	[36,37]	HK/RR	HK/RR	HK/RR	HK/RR
RstB (a)	RstA	Regulation by PhoQ/PhoP	[38]	HK/RR	HK/RR	HK/RR	HK/RR
YedV (a)	YedW	Unknown		HK/RR *	HK/RR	HK/RR	HK/RR
BaeS (a)	BaeR	Myricetin, Na tungstate, Zinc	[39]	HK/RR	HK/RR	HK/RR	HK/RR
QseC (a)	QseB	Epinephrine, Norepinephrine, AI-3	[36]	HK/RR	HK/RR	HK/RR	HK/RR
EnvZ (a)	OmpR	Osmolality, pH, CHAPS, MzrA	[37]	HK/RR	HK/RR	HK/RR	HK/RR
CpxA (a)	CpxR	CpxP, Misfolded envelope proteins (pH, osmotic stress)	[36]	HK/RR	HK/RR	HK/RR	HK/RR
BasS (a)	BasR	Indole, Fe^2+/3+^	[36,37]	HK/RR *	HK/RR	HK/RR *	HK/RR
CreC (a)	CreB	Glycolytic carbon compounds	[36]	RR *	HK/RR	HK/RR	HK/RR
ArcB (b)	ArcA	Redox	[37]	HK/RR	HK/RR	HK/RR	HK/RR
**NarL family**							
	FimZ			np	HK/RR *	HK/RR	HK/RR
NarX (a)	NarL	Nitrate iron	[37]	HK/RR	HK/RR	HK/RR	HK/RR
BarA (b)	UvrY	Formate, Acetate	[40]	HK/RR	HK/RR	HK/RR	HK/RR
NarQ (a)	NarP	Nitrate iron	[37]	HK/RR	RR	HK/RR	HK/RR
RcsC /RcsD ***	RcsB	Undecaprenyl-(pyro) phosphate, outer membrane protein RcsF	[36]	HK/RR	HK/RR	HK/RR	HK/RR
EvgS (b)	EvgA	Mildly acetic pH, Monovalent cation (Na^+^, K^+^), Redox, Indole	[37,38,41]	RR *	HK/RR	HK/RR	HK/RR
UhpB (a)	UhpA	UhpC	[36]	RR *	HK/RR	HK/RR	HK/RR
**NtrC family**							
AtoS (a)	AtoC	Acetoacetate	[36]	np	np	np	HK/RR
GlnL (a) (NtrB)	GlnG (NtrC)	2-ketoglutarate, Glutamine	[42]	HK/RR	HK/RR	HK/RR	HK/RR
ZraS (a)	ZraR	Zn^2+^, Pb^2+^	[36]	HK/RR	HK/RR	HK/RR	HK/RR
**CitT family**							
CitA (a)	CitB	Citrate	[36]	np	np	RR	HK/RR
DcuS (a)	DcuR	Malate, Oxygen	[37]	HK/RR	HK/RR	HK/RR *	HK/RR
**LytTR family**							
BtsS (a)	BtsR	Pyruvate	[43]	HK/RR	HK/RR	HK/RR	HK/RR
YpdA (a)	YpdB	Pyruvate	[44]	HK/RR	RR*	HK/RR	HK/RR
**Others**							
CheA ****	CheB CheY	Metylation of methyl–accepting chemotaxis proteins	[36]	np	HK/RR	RR *	HK/RR
				np	HK/RR	HK/RR	HK/RR
	RssB	Unknown		HK/RR	HK/RR	HK/RR	HK/RR
GlrK (a)	GlrR	Unknown		HK/RR	HK/RR *	HK/RR	HK/RR
				ModD(RR)	ModD		
				Adf1617_05396(RR)			
				Asd1617_05387(HK)			

When indicated, subscript numbers show the total number of HKs and RRs identified in the cited microorganism. HK/RR: both sensor and response regulator is present; RR: only the response regulator is present; np: not present, absence of both Sensor and Response regulator. * Indicates differences among strains of the same species (arranged from P2CS database, http://www.p2cs.org/, accessed on 1 May 2022). **: (a) and (b) indicate Type A HK (sensor) and Type B HK (hybrid sensor), respectively, as shown in Figure 1 (Class I HK); GlnL (NtrB) and CheA are cytoplasmic proteins, while the other HKs are membrane proteins. *** RcsC and RcsD are combined to form a hybrid sensor [45]. **** In CheA (Class II HK), the Hbox containing the His residue to be phosphorylated is in the HPt domain instead of the DHp domain [46].

**Table 2 biomolecules-12-01321-t002:** TCSs contributing to *Shigella* virulence.

Effect on	TCS	Gene Regulated by TCS	Targets	Ref.
**Invasion of epithelial cells**	CpxA/CpxR	*virF*, *virB*	T3SS ^a^ and its effectors	[49,50,51]
	EnvZ/OmpR	*ompC*	OmpC porin	[52,53]
	ArcB/ArcA	*iuc*, *sit* and *feo* ^b^ operons	Iron transport systems	[54]
**Survival within macrophages**	EvgS/EvgA	*emrKY*	EmrKY efflux pump	[55]
**Inflammatory response and resistance to CAMPs ^c^**	PhoQ/PhoP	*virK*, *msbB2*, membrane biosynthesis genes	Synthesis and modification of cell envelope	[56,57]

^a^ T3SS: Type 3 Secretion System; ^b^ Regulation of *feo* operon also requires Fnr regulator activity; ^c^ CAMPs -Cationic Antimicrobial Peptides.

## Data Availability

Not applicable.

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
