# Peer review of "Roles of Two-Component Signal Transduction Systems in Shigella Virulence"

_biomolecules, 2022, doi:10.3390/biom12091321_

Round 1

Reviewer 1 Report

The manuscript by Pasqua and co-workers is a very nice well-written review on the role of two-component system in controlling virulence in pathogenic organisms with a special focus on Shigella spp. The manuscript starts with a description of the virulence mechanisms that allow Shigella to invade and spread across colonic epithelial cells. Successful invasion is intimately linked to correct regulation of virulence genes in response to environmental stimuli. Shigella two-component signal transduction systems play a major role in this process. Five TCS (CpxA/CpxR, EnvZ/OmpR, EvgS/EvgA, 574 ArcB/ArcA, and PhoQ/PhoP) are central in controlling Shigella virulence and they are extensively discussed in the following chapters. The review ends with a paragraph illustrating the potential of TCS as targets for anti-virulence therapies.

Overall, the review is exhaustive, well organised, and understandable even to non-expert in the field. Few minor comments that the authors might address in the revised version of the manuscript are reported below.

Minor comments

Lines 264-276. When describing the activation mechanisms of E. coli Cpx TCS the authors might include NlpE the outer membrane lipoprotein that acts as the positive regulator sensing a subset of stimuli that activate Cpx.

Lines 537 and 540: EPs abbreviation should be defined.

Line 599: “alternate” should be “alternative”

Table 1

E. coli CheA HK is labeled with 4 asterisks (*), please include the label in the table legend when describing the specificity of this sensor.

Author Response

Lines 264-276. When describing the activation mechanisms of E. coli Cpx TCS the authors might include NlpE the outer membrane lipoprotein that acts as the positive regulator sensing a subset of stimuli that activate Cpx.

  • In agreement with the referee’s request, we added a sentence to address the role of the NlpE lipoprotein in the activation of the Cpx system (lines 292-294 rev manuscript) and the corresponding reference  Delhaye et al., 2019

Lines 537 and 540: EPs abbreviation should be defined.

  • We eliminated the abbreviation in the text

Line 599: “alternate” should be “alternative”

  • OK , done

Table 1

  1. coli CheA HK is labeled with 4 asterisks (*), please include the label in the table legend when describing the specificity of this sensor.
  •    OK, done

Reviewer 2 Report

Pasqua and colleagues have written a comprehensive review on the role of two component signalling system in the virulence of Shigella.  After a brief overview of pathogenicity mechanisms of Shigella, two-component systems are presented in general. This is followed by a detailed explanation of five selected two-component systems.  Finally, two-component systems are discussed as potential drug targets.The topic is of interest to a broad readership interested in bacterial pathogenesis and gene regulation. I have some comments that I believe may help the reader to better navigate this review. 

  line 650: please introduce the H-box  as well as the N G1 F G2 boxes in chapter 4   line 651: class I  HKs need to be introduced and compared to class II HKs in chapter 4   The manuscript contains in total only one figure (explaining the general composition of two-component systems). Schemes on functional mechanisms and effects of one or the other selected two-component system or on the response of Shigella to a specific environmental stimulus would facilitate the understanding of the text.   minor points: line 154: phosphate should start with a small letter   line 161 correct in DHp   Table 1: There is a recent report (in BioRxiv)  that YedV senses HOCl-stress   line 520: put a comma instead of a point behind gadC.

Author Response

line 650: please introduce the H-box  as well as the N G1 F G2 boxes in chapter 4  

  • In agreement with the referee’s request we have introduce H-box and N G1 F G2 boxes in chapter 4 line 170-173 (rev manuscript)

line 651: class I  HKs need to be introduced and compared to class II HKs in chapter 4

  • In agreement with the referee’s request we have introduced the concept of HK class I and class II in chapter 4 line 197-199 (rev manuscript)

The manuscript contains in total only one figure (explaining the general composition of two-component systems). …..

  • In agreement with the referee’s request we have added a new figure (Fig. 2) in order to schematically present the role played by TCSs in the different steps of Shigella invasive process

minor points: line 154: phosphate should start with a small letter  

  • OK ,done

line 161 correct in DHp

  • OK, done 

Table 1: There is a recent report (in BioRxiv) that YedV senses HOCl-stress.

  • The information is interesting but the corresponding paper is not yet published.  

line 520: put a comma instead of a point behind gadC.

  • OK, done